# Metabolic Regulation and Saline–Alkali Stress Response in Novel Symbionts of *Epichloë bromicola-Bromus inermis*

**DOI:** 10.3390/plants14071089

**Published:** 2025-04-01

**Authors:** Mengmeng Zhang, Chong Shi, Chuanzhe Wang, Yuehan Yao, Jiakun He

**Affiliations:** College of Resources and Environment, Xinjiang Agricultural University, Urumqi 830052, China; 320223517@xjau.edu.cn (M.Z.); 320223547@xjau.edu.cn (C.W.); m16310467636@163.com (Y.Y.); 320233612@xjau.edu.cn (J.H.)

**Keywords:** grass fungal endophyte, *Bromus inermis*, artificial inoculation, sterile seedling inoculation method, differential metabolite, saline–alkali stress

## Abstract

*Epichloë* endophytic fungi are important microbial resources in agriculture and animal husbandry. Because of their stable symbiosis, species transmission, and positive effects on host plants, the use of endophytic fungi in grass breeding is of great significance. In this study, six inoculation methods were used, including the sterile seedling slit inoculation method, sterile seedling cut inoculation method, sterile seedling injection inoculation method, seed soaking inoculation method, seed piercing and then soaking inoculation method, and seed slit inoculation method. Spectrometry was used to construct new symbionts, and Liquid Chromatography–mass spectrometry was used to analyze the effects of endophytic fungi on the metabolism of new hosts. The physiological response of the new symbionts to salt and alkali stress was studied using a pot experiment. The results were as follows: In this study, *Epichloë bromicola* was successfully inoculated into *Bromus inermis* via the sterile seedling slit inoculation method, and new symbionts (EI) were obtained; the vaccination rate was 2.1%. Metabolites up-regulated by EI are significantly enriched in citrate cycle and ascorbate and aldarate metabolism, suggesting that the symbiosis of endophytic fungi indirectly triggers the production of reactive oxygen species (ROS) through multiple metabolic pathways. The saline–alkali stress test showed that the host antioxidant system was active after inoculation, and the total antioxidant capacity was significantly increased compared with non-symbionts (EF) under mild stress (*p* < 0.05), which provided important clues to reveal the complex mechanism of plant–fungus symbiosis. This study provides practical guidance and a theoretical basis for plant adaptation under climate change, health management of grass seeds, and soil improvement through endophytic fungi.

## 1. Introduction

*Epichloë* is a grass endophytic fungus that has been extensively studied for its wide distribution in the grass family and its ability to increase the systemic resistance of the host plant to biotic and abiotic stresses [1,2]. It is worth noting that *Epichloë* endophytic fungi mainly spread through the seeds of the host [3], which makes the symbiotic relationship very stable [4,5] but also results in the naturally formed “one-to-one” relationship between different *Epichloë* strains and the host. Because of the species transmission characteristics of *Epichloë* endophyte and its positive effect on host plants, researchers began to try to construct artificial grass–endophyte unnatural symbionts [6,7], called “new symbionts”, on the one hand, to break through the host specificity and study symbiosis [1,4]. On the other hand, it is of great significance for improving agricultural production efficiency and ecological sustainability to construct artificial symbionts between excellent strains and hosts with excellent traits.

However, the practice of artificial inoculation is facing certain difficulties. Due to the host specificity of endophytic fungi, the high operational requirements of the artificial inoculation process, and the need to destroy plant tissues, the host mortality is high, leading to the low success rate of artificial inoculation. In the 1980s, Latch et al. first used the sterile seedling slit inoculation method to directly access the mycelium by cutting the stem of a seedling horizontally or vertically. The total vaccination success rate was 10.8% [8]. Wang et al. introduced bacterial suspension into the plant using seed soaking and injection [9]. Although this method facilitates mycelia invasion, the seed structure is destroyed, normal seed growth and development are affected, and the seed survival rate is reduced, thus preventing mycelia from entering the seedlings during seed germination and growth. The selection of inoculation methods, genotypes of endophytic fungi, types of inoculation tissues and subsequent culture conditions had a great influence on the success rate of inoculation [7,10]. However, with the continuous discovery and excavation of new endophytic fungi resources, the space of using endophytic fungi for breeding is gradually expanding. In recent years, especially in the field of pratacultural science, some new varieties of turf grass and forage have been successfully cultivated. New Zealand has developed a number of commercial strains of *Epichloë festucae*, such as MaxQ^®^ and MaxP^®^ strains, which are widely used in *Festuca arundinacea*, giving the new germplasm excellent pest resistance [11,12]. The AR series and MEA series strains gave *Lolium perenne* stronger insect resistance and enhanced the adaptability of the grass species [13,14]. In China, Li et al. artificially inoculated the endophytic fungus (*Epichloë bromicola*) from *Hordeum brevisubulatum* into the closely related *Hordeum vulgare*, significantly increasing the production of neosymbionts [15]. These few successful cases, mainly using endophytic fungi to improve insect resistance or a high yield of new hosts [11], have not yet constructed a new symbiont to fully explore its potential for environmental adaptation. The use of endophytic fungi to improve the environmental adaptability of grass seeds, especially the ability to grow in extreme environments, may play an important role in the health management of grass seeds and soil protection.

*Bromus inermis* is a perennial grass, widely distributed in Xinjiang, China. Its roots are very developed, and it has the characteristics of drought tolerance, salt and alkali tolerance, and wide adaptability. It is usually used to improve and utilize saline–alkali land and gradually reduce the salt content in soil through the absorption and accumulation of salt during growth so as to improve soil structure and fertility [16,17]. *B. inermis* almost does not carry endophytic fungi under natural conditions. The purpose of this study was (1) to determine the inoculation method of the new symbionts artificially constructed by *Epichloë-B. inermis*; (2) differences in metabolism between new symbionts and non-symbionts; and (3) adaptability of new symbionts to saline–alkali environments.

## 2. Results

### 2.1. Artificial Inoculation Result

The endophytic fungus *E. bromicola* was successfully inserted into *B. inermis* sterile seedling using the sterile seedling slit inoculation method (Figure 1). A total of 1455 sterile seedling strains were inoculated with this method, and 1136 survived after inoculation. The results of the microscope examination and re-isolation culture showed that 24 strains were successfully inoculated, with a success rate of 2.1% (Table 1). The mycelium grew parallel to the cell wall in successfully inoculated plants, and the endophytic fungal colonies obtained by re-isolation were consistent with *E. bromicola* colonies. Three endophytic fungi were artificially inoculated into *B. inermis* using six methods. Detailed data are in shown Appendix A.

### 2.2. Effects of E. bromicola on Metabolism of B. inermis in a New Host

#### 2.2.1. Principal Component Analysis

PCA was used to compare the overall differences in metabolites between the EI and EF groups to determine whether *E. bromicola* had an effect on the host metabolism. Figure 2 shows the PCA results of the two groups of samples in positive and negative ion modes, respectively. The total interpretability shown in the two figures is 65.60% and 60.19%, respectively, which can better show the clustering of samples. Although there was some overlap between the two groups, they still showed obvious differences, indicating that *E. bromicola* had a significant effect on the metabolism of the new host.

#### 2.2.2. Screening of Differential Metabolites

The VIP value was calculated by OPLS-DA, and the differential multiple was used to screen the differential metabolites between the two groups. Figure 3 shows that in both positive and negative ion modes, the R^2^Y of the model is greater than 0.9, and Q^2^ is greater than 0.5, indicating that the model has a good distinction effect and no overfitting phenomenon.

We screened differential metabolites by VIP > 1 and FC > 1.5. Figure 4 shows the differential metabolites of EI vs. EF in positive and negative ion modes. The up-regulated metabolites in the EI group were N2, n2-dimethylguanosine, sinapyl alcohol, and fucoxanthin. The up-regulation of these metabolites suggests that endophytic fungi may up-regulate the antioxidant capacity, stress resistance, and defense metabolism of plants by regulating their metabolic pathways [18,19]. In contrast, down-regulated metabolites include chlorhexidine, pro-thr, and 9-ketofluprosternol isopropyl ester. These down-regulated metabolites may be related to plant immune response, nutrient metabolism, or growth regulation, suggesting that plants lacking endophytic fungi are not effectively activated or maintained in these metabolic pathways [20]. In summary, the presence of endophytic fungi significantly affected the levels of metabolites in plants, and plants carrying endophytic fungi showed up-regulation in several metabolites, which may be related to plant stress resistance, adaptability, and defense metabolism.

#### 2.2.3. KEGG Pathway Enrichment Analysis of Differential Metabolites

Figure 5A shows the up-regulated differential metabolite enrichment analysis in the EI group compared to the EF group. The results showed that pathways such as energy metabolism (citrate cycle) and antioxidant metabolism (ascorbate and aldarate metabolism) were significantly enriched. As the core of energy metabolism, the citrate cycle may play an important role in the symbiosis between plants and endophytic fungi, enhancing the energy supply and metabolic adaptability of plants. The up-regulation of ascorbate and aldarate metabolism may indicate that the antioxidant defense mechanism of plants has been strengthened after fungal invasion [21], which helps to reduce the oxidative stress caused by fungi. Figure 5B shows the enrichment results of down-regulated differential metabolites in the EI group compared to the EF group. Glycerolipid metabolism and glyoxylate and dicarboxylate metabolism pathways were significantly enriched, and glycerolipid metabolism is closely related to cell membrane synthesis and energy storage in plants [22]. This suggests that the infection of endophytic fungi inhibits lipid synthesis in the host, while glyoxylate and dicarboxylate metabolism reflect the inhibition of carbon utilization in plants.

### 2.3. Physiological Response of Epichloë bromicola-B. inermis to Saline–Alkali Stress

Figure 6 shows the physiological indexes of EI and EF plants under salt–alkali stress. The results show that the APX, GR, T-AOC, and soluble sugars of EI plants are significantly higher than EF under light salt–alkali stress (*p* < 0.05). The GR, T-AOC, soluble sugars, pro, and MDA of EI plants were significantly lower than those of EF plants under moderate saline–alkali stress (MSA) (*p* < 0.05). The APX, GR, T-AOC, soluble sugars, and pro of EI plants were significantly lower than those of EF plants under severe stress (*p* < 0.05). Salt and alkali stress activated the antioxidant enzymes and total antioxidant capacity of plants, reduced the membrane lipid peroxidation damage, and enhanced the adaptability of plants to salt and alkali stress through metabolic adjustment [23]. With the deepening of salt and alkali stress, the response mechanism of plants to salt and alkali stress was gradually strengthened.

Analysis of ion concentrations in EI and EF plants showed (Figure 7) that Ca^2+^ and K^+^ in EI plants were significantly lower than EF in the MSA condition (*p* < 0.05). In HSA, Ca^2+^, K^+^ and Na^+^ were significantly lower than EF (*p* < 0.05). When salt–alkali stress reaches a certain extent, it may affect the overall growth of plants.

## 3. Discussion

### 3.1. Research on Artificial Inoculation Methods

At present, the artificial inoculation methods for grasses are mainly sterile seedlings, callus, seeds and adult plants as inoculation materials, and the sterile seedling inoculation method is the most commonly used inoculation method at present [8,9,24,25]. Its operation is simple and efficient, and it is suitable for large-scale inoculation. The disadvantage is that it needs to cause wounds to seedlings, which can easily lead to contamination or death. At present, the mainstream method of successful inoculation is sterile seedling inoculation, and this study is no exception. Internationally, some scholars used *Lolium perenne*, *Festuca arundinacea* and *Festuca rubra* as research objects to inoculate *Epichloë festucae* via the sterile seedling inoculation method, with an average success rate of 10.8% [8]. Chen et al. introduced *Epichloë gansuensis* into *Achnatherum inebrians* sterile seedlings via the sterile seedling inoculation method, and the inoculation rate was 21% [26]. The endophytic fungi of *Epichloë* existed in the above two research objects under natural conditions, and the endophytic fungi in the plant were isolated and then grafted back, so the vaccination rate was relatively high. Li et al. added *E. bromicola* to *Hordeum vulgare* sterile seedling, and the vaccination rate was 2% [15]. In this study, the endophytic fungus *E. bromicola* was inoculated into *B. inermis* via the sterile seedling slit inoculation method, and the success rate was 2.1%. The subjects studied by Li et al. did not have *Epichloë* endophytic fungi under natural conditions, and the vaccination rate was similar to that in this study. All strains successfully inoculated were *E. bromicola*. This may be related to the wide existence of *E. bromicola* in natural conditions and its rich host species in *Bromus*, *Leymus*, *Hordelymus*, *Roegneria*, *Agropyron*, *Elymus* and other plants, indicating that *E. bromicola* has good host compatibility [27,28]. The results of artificial inoculation experiments indicate that endophytic fungi with a wide range of hosts are more likely to be successfully inoculated, which complements the *Epichloë* inoculation principle.

### 3.2. Effects of E. bromicola on Metabolism of B. inermis in a New Host

The metabolites in EI plants were significantly upregulated, including apigenin 7,4′-dimethyl ether, mandelic acid methyl ester, mevalonic acid and vitamin C. These compounds are closely related to plant stress resistance, antioxidant capacity, and growth regulation, in particular, the up-regulation of Vitamin C, a powerful antioxidant that is normally involved in the process of removing reactive oxygen species [29,30]. Plants typically respond to ROS production by increasing the synthesis of antioxidants when they are stressed or infected by pathogens [31], so the up-regulation of Vitamin C may indicate that plants are experiencing some level of oxidative stress in their symbiotic relationship with endophytic fungi. In addition, the enrichment of the citrate cycle (TCA cycle) pathway also provides support for this hypothesis. The TCA cycle is a central pathway of the cellular metabolism, and, in addition to playing an important role in energy production, it also produces ROS in plant respiration. The symbiosis of endophytic fungi may further promote ROS production by regulating the activity of the TCA cycle, changing the metabolic balance of plant cells. This process may be a response mechanism of plants to fungal symbiotic stimulation [32]; in this process, plants improve energy supply by activating metabolic pathways such as the TCA cycle, but it may also bring oxidative stress [33,34]. In addition to oxidative stress, metabolites associated with lipid metabolism and amino acid metabolism were significantly down-regulated in EI plants, especially glycerolipid metabolism, glyoxylate and dicarboxylate metabolism. It may suggest that endophytic fungi alter energy storage and cell membrane construction in plants by regulating these metabolic pathways [35]. The down-regulation of glycine, serine and threonine metabolism may affect protein synthesis and metabolic balance in plants, and these changes may be a way for plants to optimize resource allocation during symbiosis [36]. In summary, the symbiosis of endophytic fungi may indirectly trigger the production of reactive oxygen species (ROS) in plants through multiple metabolic pathways, thus activating the antioxidant defense mechanism of plants [37,38]. This mechanism provides a new perspective for metabolic regulation in plant adaptation to symbiosis and suggests that we should further study the oxidative stress response in endofungus–plant symbiosis and provide important clues for revealing the complex mechanism of plant–fungus symbiosis.

### 3.3. Physiological Response of Symbiont to Saline–Alkali Stress

Saline–alkali stress is one of the major abiotic stresses in global agricultural production, especially in arid and semi-arid areas, where soil salinization severely limits crop growth and yield. It is of great ecological and agronomic significance to improve the saline–alkali tolerance of plants by constructing new symbionts and using endophytic fungi. When plants are subjected to salt and alkali stress, they will accumulate too much ROS [39], leading to oxidative stress and membrane lipid peroxidation, further damaging plant tissues. Plants have their own antioxidant defense system, which can effectively remove excess ROS [40], alleviate plant damage, and increase plant stress resistance. The synthesis of osmoregulatory substances is one of the important ways for plants to adapt to stressed environments. The organic solvents synthesized by plants mainly include proline and soluble sugar. Studies have shown that *Epichloë* endophytic fungi symbiotic with grasses under natural conditions can improve the resistance of plants to salt–alkali stress [41], and many studies have proved that *Bromus inermis* itself has a good salt–alkali tolerance [17]. In this study, EI plants showed a high sensitivity under mild salt–alkali stress. GR, T-AOC, soluble sugars, pro, and MDA showed different decreasing trends, while the APX, GR, T-AOC, and soluble sugars of EF plants reached their peak values at moderate stress, indicating that EI plants were more sensitive to saline–alkali stress. Under severe salt–alkali stress (HSA), the APX, GR, T-AOC, soluble sugars, and pro of EI plants are significantly lower than EF, indicating that the antioxidant capacity, organic solvent synthesis, and oxidative stress relief ability of EI plants are decreased with an increase in salt–alkali stress [42]. Na^+^, K^+^, Ca^2+^ and other inorganic salt solutes are another mechanism of osmoregulation in plants [43]. In this study, the K^+^ and Ca^2+^ concentrations of EI plants were gradually lower than EF, and Na^+^ was gradually higher than EF as the strain severity increased. In addition, the K^+^ concentration of EI plants was significantly lower than that of EF under MSA and HSA. The accumulation of K^+^ in plant cells under stress is crucial for stomatal conductance, and a decrease in the K^+^ concentration may hinder the normal activities of plants. The antioxidant capacity of EI plants treated without saline–alkali stress (CK) was significantly reduced, which was consistent with the results of metabolomics analysis. Chen et al. studied the response of wild barley (*Hordeum brevisubulatum*) containing *Epichloë* endophytic fungi to salt and alkali stress and found that the presence of endophytic fungi could promote plant growth under salt and alkali stress, which was contrary to the results of this study [41]. Combined with the metabolic analysis of plants in this study, this may be due to the fact that more energy and materials are used to resist the invasion of endophytic fungi, which have some incomplete compatibility with the host.

In this study, only three stress gradients were set in the saline–alkali stress test, which failed to cover a wider range of stress conditions, which may have affected the comprehensive assessment of symbiont adaptability. Future studies can further validate and extend the results of this study by increasing the optimization of inoculation methods, expanding the combination range of strains and hosts, and setting a wider range of stress conditions.

In the future, through further research on the interaction mechanism between host and endophytic fungi, combined with molecular breeding and gene editing technology, further research and technological innovation can be carried out. Endophytic fungi are not only expected to improve the pest resistance, yield, and stress resistance of grass seeds but may also play an important role in the health management of grass seeds and soil protection. The use of endophytic fungi for grass seed improvement has broad prospects, which provides a new solution for the sustainable development of global agricultural production, especially the grass industry.

## 4. Materials and Methods

### 4.1. Strain and Plant Origin

The seeds of *B. inermis cv.* Wusu No. 1 were provided by the Pratacultural College of Xinjiang Agricultural University. The three strains used in this study were *Epichloë Bromicola* isolated from *Elymus dahuricus*, which could form symbionts with a variety of grasses and improve the stress resistance of the original host [27]. *Epichloë guerinii*, isolated from *Melica*, is widely distributed in Europe and Central Asia, but in China, it is only naturally distributed in the north slope of Tianshan Mountains, and it is a strain with local adaptability [44]. *Epichloë elymi* was isolated from *Bromus japonica*, and its natural host was closely related to *B. inermis*.

### 4.2. Artificial Inoculation Method

Mature seeds of *B. inermis* with uniform color and size were selected; 50 seeds were randomly selected for staining with the Bengal red method, and then the seeds were determined to be bacterially free via microscopy (Leica light microscope, 40× or 50×). The seeds were disinfected with 75% ethanol solution (Qingdao Hainuo Biological Engineering Co., Ltd., Qingdao, China) for 3 min, 5% NaClO solution (Guangdong Kena Chemical Reagent Co., Ltd., Dongguan, China) for 5 min, washed with sterile water 3–5 times, then disinfected with 75% ethanol solution for 3 min, washed with sterile water 3–5 times, and the treated seeds were placed on sterile filter paper to absorb surface water. The sterilized seeds were placed in 1/2 MS medium (Hangzhou Baisi biotechnology Co., Ltd., Hangzhou, China) and cultured in a constant temperature incubator (NingBoJiangNanYiQiChang, Jinan, China) at 23 °C for 48 h in darkness, then changed to dark culture for 12 h, light culture (light intensity of 60 μmol/(m^2^·s)) for 12 h, and continued culture for 21 days to obtain sterile seedlings. The manual inoculation methods are shown in Table 2.

The sterile seedling was cultured in an incubator for 7 days after inoculation and transplanted. Seeds were inoculated and cultured in 1/2 MS medium in an incubator 14 days before transplanting. Transplanting means that the treated plants are transferred into the sterilized substrate (volume ratio of vermiculite to nutrient soil = 1:1), cultured in an artificial climate chamber (23 ± 2 °C), and watered as needed. The control group was treated the same except inoculated with endophytic fungi, during which the survival rate of seedlings was counted.Survival rate (%) = Survival quantity/Inoculation quantity × 100%

### 4.3. Detection of Fungal Infection Rate in B. inermis

After transplanting, the plants grew for 45 days, and the culms of the plants were detected by the tissue staining method and PDA plate separation and culture method, respectively. The endophytic fungal infection in the plants was marked, and the successfully infected plants were classified as symbionts (EI), while the unsuccessfully infected plants were classified as non-symbionts (EF), and the infection rate was calculated [15]. The leaves of EI and EF plants were frozen with liquid nitrogen and sent to Wekemo Tech Group Co., Ltd. (Shenzhen, China) for the detection of metabolites using LC-MS.Infection rate (%) = Number of infections/Survival quantity × 100%

### 4.4. LC-MS Detection of Symbiont and Non-Symbiont

After thawing plant leaf samples slowly at 4 °C, 1 g fresh leaf samples was added to pre-cooled methanol/acetonitrile/aqueous solution (2:2:1, *v*/*v*), with ultrasonic vortex mixing at low temperature for 30 min, standing at −20 °C for 10 min, centrifuging at 14,000× *g* and 4 °C for 20 min, vacuum drying with supernatant, adding 100 μL acetonitrile solution during mass spectrometry (acetonitrile:Water = 1:1, *v*/*v*), redissolved, swirled, centrifuged at 14,000× *g* at 4 °C for 15 min, and the supernatant was taken for analysis. The samples were separated by Agilent 1290 Infinity LC (Thermo, Waltham, MA, USA) HILIC column according to the standards of Wekemo Tech Group Co., Ltd. (Shenzhen, China). Column temperature: 25 °C; flow rate: 0.5 mL/min; sample size: 2 μL. The samples were placed in a 4 °C automatic injector during the entire analysis process. In order to avoid the influence caused by the fluctuation in the instrument detection signal, the samples are analyzed continuously in random order. Quality control (QC) samples are inserted into the sample queue to monitor and evaluate the stability of the system and the reliability of the experimental data. After the samples were separated by LC, mass spectrometry was performed using the Triple TOF 6600 mass spectrometer (AB SCIEX, Framingham, MA, USA), and the samples were collected by electrospray ionization (ESI) positive ion (Pos) and negative ion (Neg) modes.

### 4.5. Salt and Alkali Stress Test of Symbiont

A pot experiment was conducted in an artificial climate chamber, and 1/2 strength Hoagland’s with salt and alkali (molar ratio of NaCl:NaHCO_3_ = 1:1) was used to treat EI and EF plants (reagents were from Hangzhou Baisi biotechnology Co., Ltd., Hangzhou, China), with Na^+^ concentration as the standard, and 4 concentrations were set: CK group (0); mild saline–alkali LSA group (100 mmol/L); moderate saline MSA group (200 mmol/L). In the severe saline–alkali HSA group (300 mmol/L), there were 8 treatment groups with 3 replicates in each group and 24 POTS with a volume of 500 mL each. Further, 150 mL of salt and alkali solution was added to each basin every 7 days, and the liquid that permeated into the tray was poured back into the basin. The first salt–alkali solution was poured on the first day of the test, and the test period was 21 days. When the coercion was over, Nanjing Aoqing Biotechnology Co., Ltd. (Nanjing, China) APX (Ascorbate peroxidase), GR (Glutathione Reductase), T-AOC (Total Antioxidant) produced by China Capacity, Pro (Proline), malondialdehyde (MDA), soluble sugars assay kit, and relevant parameters were determined according to the instructions. The dried sample was broken into powder, heated, and digested at 180 °C for 60 min, and then introduced by graphite oxide furnace using the Agilent 5100 ICP-OES system (Agilent Technologies, Santa Clara, CA, USA). The plasma temperature is set at 8000–10,000 K and appropriate wavelengths are used for the detection of sodium (Na^+^, 589.0 nm), potassium (K^+^, 766.5 nm) and calcium (Ca^2+^, 423.3 nm) ions.

### 4.6. Data Visualization and Statistical Analysis

The original data detected by LC-MS were converted into .mzXML format by ProteoWizard (v 3.0), and then XCMS (v 1.41.0) software was used for peak alignment, retention time correction, and peak area extraction. The metabolite structure of the data extracted by XCMS was first identified, and then the metabolite content was standardized for correction (in-sample correction was first performed; that is, the abundance of all features in the sample was divided by the median abundance of the sample; then the abundance matrix correction was carried out; that is, log conversion was performed for all the abundance values). Finally, the internal feature correction was carried out; that is, the abundance of all samples corresponding to the feature is subtracted by the mean abundance of the feature and then divided by the standard deviation of the abundance of the feature. The final metabolite content table for analysis was obtained. Principal Component Analysis (PCA) and OPLS-DA (Orthogonal Partial Least Squares Discriminant Analysis) were performed using R (version 3.5.1). The Variable Importance in Projection (VIP) and Fold Change (FC) values of each metabolite were calculated. Based on an analysis of the site (MetaboAnalyst 5.0: https://www.metaboanalyst.ca, accessed on 10 January 2025), a metabolism path analysis on different metabolites was carried out. The data of plant enzyme activity and ion content were analyzed by one-way ANOVA using GraphPad Prism (V 9.3.0) and plotted.

## 5. Conclusions

In this study, a new symbiont of *Epichloë bromicola-B. inermis* was successfully constructed using the sterile seedling slit inoculation method, and the vaccination rate was 2.1%. Metabolic studies of neosymbionts suggest that the symbiosis of endophytes indirectly induces reactive oxygen species (ROS) in plants through multiple metabolic pathways and inhibits lipid synthesis and the carbon source utilization of the host. The saline–alkali stress test showed that the host antioxidant system was active after inoculation, and the total antioxidant capacity was significantly increased compared with that of non-symbionts (EF) under mild stress (*p* < 0.05). However, with an increase in saline–alkali stress, the antioxidant capacity and osmotic regulation ability of the symbionts decreased gradually, indicating that the adaptability of the symbionts in extreme saline–alkali environments has certain limitations.

## Figures and Tables

**Figure 1 plants-14-01089-f001:**
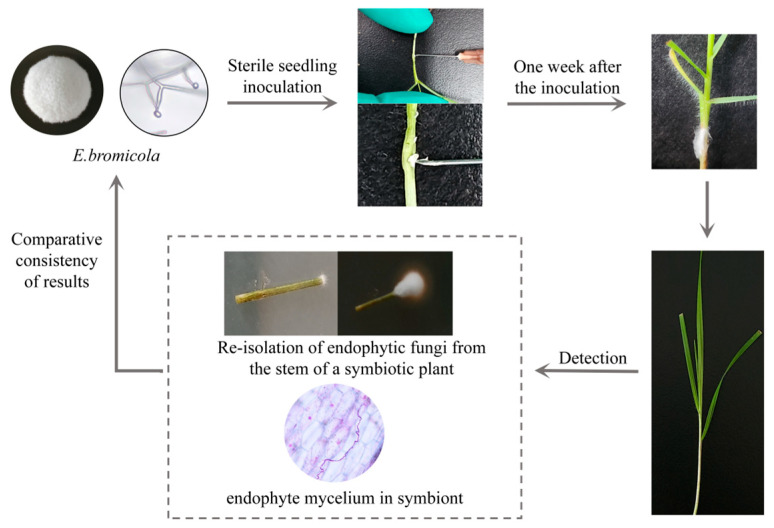
Artificial construction of *E. bromicola-B. inermis* symbiont.

**Figure 2 plants-14-01089-f002:**
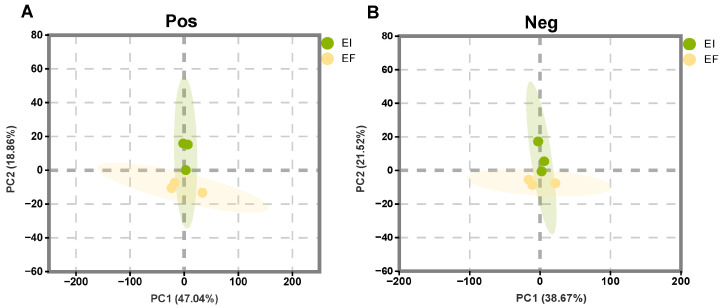
PCA scores of metabolites in EI and EF groups. Positive ion mode (**A**); negative ion mode (**B**).

**Figure 3 plants-14-01089-f003:**
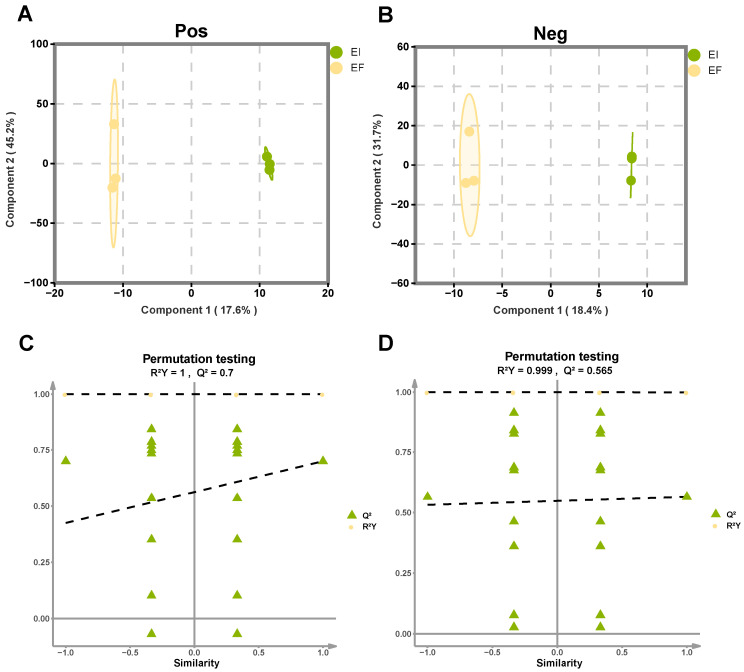
OPLS-DA score map and arrangement test results of metabolites in EI and EF groups. Positive ion mode (**A**,**C**); negative ion mode (**B**,**D**).

**Figure 4 plants-14-01089-f004:**
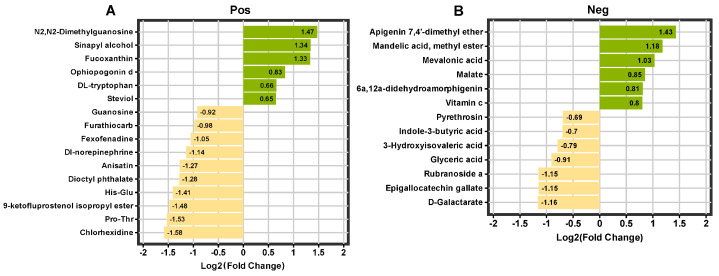
Differential multiples of differential metabolites. Positive ion mode (**A**); negative ion mode (**B**).

**Figure 5 plants-14-01089-f005:**
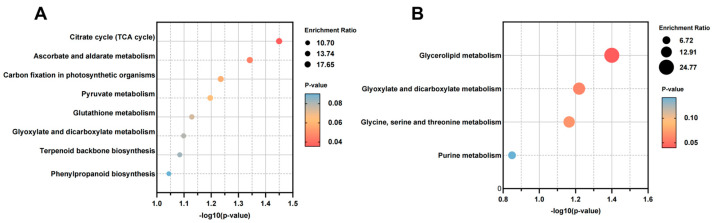
Enrichment analysis of differential metabolites of up-regulated (**A**) and down-regulated (**B**) EI vs. EF.

**Figure 6 plants-14-01089-f006:**
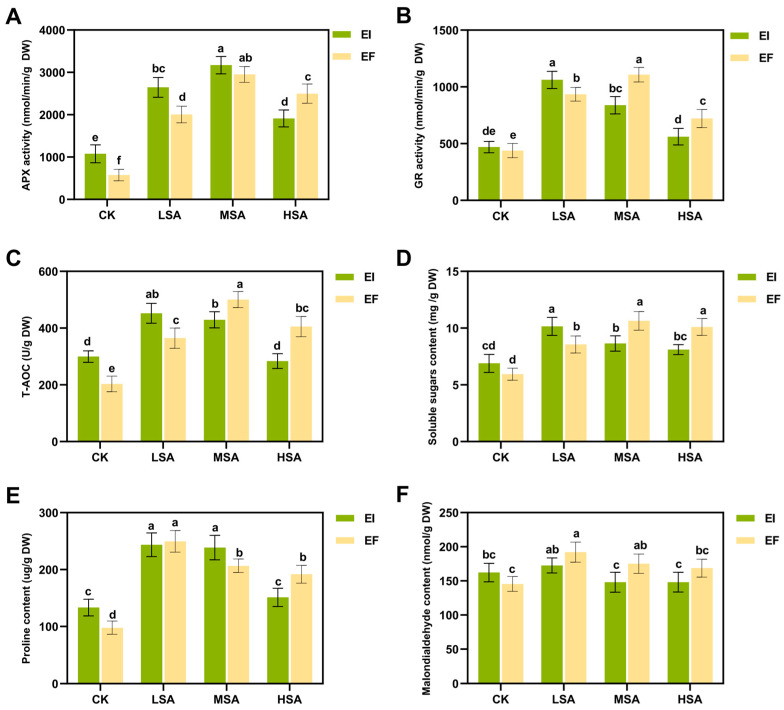
Physiological indices of EI and EF plants under saline–alkali stress. APX activity (**A**), GR activity (**B**), T-AOC (**C**), soluble sugars content (**D**), proline content (**E**), malondialdehyde content (**F**). Different lowercase letters indicate significant differences *p* < 0.05.

**Figure 7 plants-14-01089-f007:**
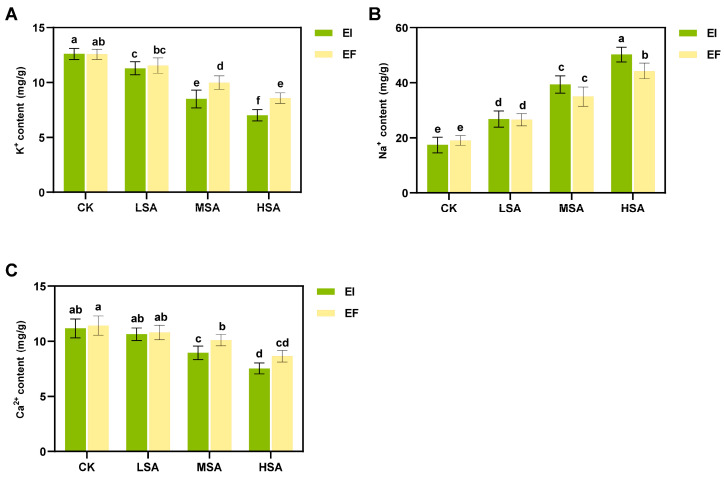
Contents of K^+^ (**A**), Na^+^ (**B**) and Ca^2+^ (**C**) in EI and EF plants under salt and alkali stress. Different lowercase letters indicate significant differences *p* < 0.05.

**Table 1 plants-14-01089-t001:** Artificial inoculation result.

Method	Fungi	Number of Inoculated Seedlings	Seedling Survival Rate (%)	Fungal Infection Rate (%)
sterile seedling slit inoculation method	*E. bromicola*	1455	78.1	2.1
	CK	427	80.6	0

**Table 2 plants-14-01089-t002:** Artificial inoculation method for constructing *Epichloë-B. inermis* symbiont.

Material	Method	
Sterile seedling	Cut	Cut the sterile seedling 2 cm above ground with sterile scissors and cover the wound with inoculated mycelium.
	Slit	A 2–3 mm incision was made at the meristem of a sterile seedling with a sterile scalpel, and hypha was inserted into the wound.
	Injection	The spore suspension was injected into the meristem of sterile seedlings.
Seeds	Soaking	The disinfected seeds were placed in a petri dish with bacterial suspension and soaked for 24 h.
	Piercing and soaking	The sterilized seeds were pierced near the embryo and then sealed and soaked in bacterial suspension for 24 h.
	Slit	The sterilized seeds are cut into small holes near the embryo and the mycelium is inserted into the wound.

## Data Availability

Data are contained within the article and Appendix A.

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
