# Peer review of "Metabolic Regulation and Saline–Alkali Stress Response in Novel Symbionts of Epichloë bromicola-Bromus inermis"

_plants, 2025, doi:10.3390/plants14071089_

Round 1
Reviewer 1 Report
Comments and Suggestions for Authors
The article presents an interesting novel study. In my opinion it needs a few clarifications/ improvements before it can be considered for publication.
The study investigates the artificial inoculation of Epichloë bromicola into Bromus inermis and how this symbiosis affects metabolic pathways and physiological responses to saline-alkali stress. The study is original and relevant as it explores relatively unexplored areas in plant-microbe symbiosis and stress adaptation research. It extends prior research by demonstrating the metabolic and physiological effects of E. bromicola in a new host species, providing insights into how artificial inoculation could be used for stress-resistant grass development.
Methodology improvements are mentioned below (specific lines mentioned).
The conclusions are consistent with the results.
References are appropriate. An additional reference position is mentioned below.
The tables and figures are clear and easy to understand and do not need to be improved.
Improvements:
Introduction: The aim of the study, especially its novelty compared to the research so far, could be more emphasized.
Results: Line 89: Add more details on the microscope examination (perhaps in Materials and Methods), what microscope was used, and at which magnification.
Materials and Methods: Line 277: Codes for strains from the collection could be added if available. Were there any reasons to choose those particular strains?
Line 291: Please give the producers of this and other media used in your experiments (check the rest of the manuscript)
Line 311: Include a reference for the infection rate calculation
Comments on the Quality of English LanguageThe language is good, but there are some grammar issues and occasional extra spaces. Some sentences are a bit long and difficult to follow. The article should be rechecked, perhaps by a native English speaker or a scientific editing service.
Author Response
The detailed reply is in the attachment, please check.

Reviewer 2 Report
Comments and Suggestions for Authors
Critical manuscript evaluation report
Manuscript Title
Construction of a Novel Symbiont of Epichlo bromicola - Bromus inermis and its Metabolic Regulation and Response to Saline Alkali Stress
1. General Comments
The manuscript addresses a relevant issue in the symbiotic interaction between Epichlo bromicola and Bromus inermis, evaluating their metabolic regulation and response to saline-alkaline stress. Research has potential for the Plants journal, but requires improvements in structure, methodological clarity, and data presentation before publication.
2. Detailed comments by Section
Title
It is descriptive, but could be more concise without losing clarity. It is recommended to better specify the focus on metabolic regulation to highlight the novelty of the study.
Abstract
It presents the objectives, methodology, and main results, but should include key quantitative information to strengthen interpretation. Better structuring of the results with specific values and statistical significance is recommended.
Introduction
It contextualises the relevance of the study well, but needs a clearer justification for the knowledge gap. The references used should be updated with recent literature (the last 5 years) to better support the theoretical framework. Include a clearer statement of the study's specific hypothesis and objectives.
Materials and Methods
More detail on the growing and experimental conditions is needed to ensure the reproducibility of the study. The statistical analysis is not sufficiently described. It is essential to add post hoc tests and coefficients of variation. There is a lack of information on experimental controls and validation of the data obtained.
Results
Figures and tables are informative, but they need a better presentation and more detailed descriptions. It is important to include values of statistical significance and precision in comparisons between groups. The organisation of the results could be improved to make the narrative of the study clearer.
Discussion
Comparisons are presented with previous studies, but the explanation of the underlying mechanisms needs to be strengthened. It would be advisable to broaden the discussion on the ecological and agronomic implications of symbiosis in saline-alkaline conditions. The limitations of the study should be explicitly stated.
Conclusions
It summarises the findings well but should avoid extrapolations without experimental support. It is recommended to emphasise the impact of the study on understanding Epichlo-Bromus symbiosis under adverse conditions.
References
Several references are outdated. We recommend including recent studies. Verify that the format complies with Plants standards.
3. General recommendations
Re-visit the wording and improve the clarity of some paragraphs. Include additional methodological details to ensure reproducibility. Adjust the figures and tables to make them more understandable and complete. Strengthen statistical analysis with additional testing.
4. Strengths and weaknesses
Strengths: Relevant topic on symbolic interactions and resistance to environmental stressors. Use of appropriate experimental methodology. Potential contribution to biotechnology and plant ecology.
Weaknesses: Lack of methodological and statistical details. Some sections require greater clarity and structure. Outdated references.
5. Eligibility for publication
Publication is recommended after further review, ensuring that the authors address the following aspects: Clarify the methodology and statistical analysis. Update references with recent studies. Optimise figures and tables for clarity. Improve the organisation and coherence of the text.
If these revisions are implemented, the manuscript will be a strong candidate for publication in Plants.
Reviewer, 2025.02.25
Comments on the Quality of English LanguageThe English could be improved to more clearly express the research.
Author Response

(The authors gave the same response as above.)

Round 2
Reviewer 2 Report
Comments and Suggestions for Authors
Manuscript Evaluation Report (Version 2)
Title: Metabolic regulation and saline-alkali stress response in new symbionts of Epichlo bromicola-Bromus inermis
1. Strengths of the new version
- The title has been appropriately adjusted as per the previous recommendation, is now concise, and clearly reflects the focus on metabolic regulation.
- The abstract section now presents clear quantitative information and statistical significance values, strengthening interpretation, and highlighting relevant aspects of the study.
- The introduction was considerably improved by updating the references with recent literature (last 5 years), clearly justifying the knowledge gap, and explicitly formulating the hypothesis and specific objectives of the study.
- The Materials and Methods section now includes a more detailed description of the experimental conditions and statistical analyses, including additional methods such as post hoc tests, increasing the reproducibility of the experiment.
- The presentation and description of the results were improved with clearer tables and figures, with explicit indications of statistical significance, facilitating the understanding of the findings.
- The discussion was significantly strengthened by delving into the underlying metabolic mechanisms and broadening the reflection on the ecological and agronomic implications of the study, as well as clearly mentioning the limitations of the study.
- The conclusions have been adjusted to avoid extrapolations without experimental support, clearly highlighting the contribution of the study to understanding the Epichlo-Bromus symbiosis under saline-alkaline stress.
- The linguistic quality and writing in English improved markedly, making it easier to read and understand the manuscript.
2. Additional observations or recommendations
- Although the graphical presentation improved significantly, some figures could benefit from additional adjustments in their visual clarity (e.g., size of legends and axes, choice of colours with better contrast).
- It is recommended to briefly consider possible molecular mechanisms in future studies to further expand our understanding of symbiotic interaction.
- Although the references were updated, it is important to verify again that they all strictly comply with the format required by the Plants journal Plants.
3. Conclusion and final recommendation
In this second version, the authors have satisfactorily addressed most of the observations made in the initial assessment. The modifications have substantially improved the scientific quality, clarity, and structure of the manuscript. Therefore, I recommend its acceptance for publication after the minor revision indicated above (visual clarity in figures and confirmation of the bibliographic format), without the need for a complete new revision.
Author Response

(The authors gave the same response as above.)
